# Multi-Omics Integrative Analysis to Reveal the Impacts of *Shewanella algae* on the Development and Lifespan of Marine Nematode *Litoditis marina*

**DOI:** 10.3390/ijms25169111

**Published:** 2024-08-22

**Authors:** Yiming Xue, Beining Xue, Liusuo Zhang

**Affiliations:** 1CAS and Shandong Province Key Laboratory of Experimental Marine Biology, Institute of Oceanology, Chinese Academy of Sciences, Qingdao 266071, China; xueyiming@qdio.ac.cn (Y.X.); xuebeining18@mails.ucas.edu.cn (B.X.); 2Laboratory for Marine Biology and Biotechnology, Qingdao Marine Science and Technology Center, Qingdao 266237, China; 3Center for Ocean Mega-Science, Chinese Academy of Sciences, 7 Nanhai Road, Qingdao 266071, China; 4University of Chinese Academy of Sciences, Beijing 100049, China

**Keywords:** *Litoditis marina*, development, lifespan, *Shewanella algae*, *Escherichia coli* OP50, transcriptomics, metabolomics

## Abstract

Understanding how habitat bacteria affect animal development, reproduction, and aging is essential for deciphering animal biology. Our recent study showed that *Shewanella algae* impaired *Litoditis marina* development and lifespan, compared with *Escherichia coli* OP50 feeding; however, the underlying mechanisms remain unclear. Here, multi-omics approaches, including the transcriptome of both *L. marina* and bacteria, as well as the comparative bacterial metabolome, were utilized to investigate how bacterial food affects animal fitness and physiology. We found that genes related to iron ion binding and oxidoreductase activity pathways, such as *agmo-1*, *cdo-1*, *haao-1*, and *tdo-2*, were significantly upregulated in *L. marina* grown on *S. algae*, while extracellular structural components-related genes were significantly downregulated. Next, we observed that bacterial genes belonging to amino acid metabolism and ubiquinol-8 biosynthesis were repressed, while virulence genes were significantly elevated in *S. algae*. Furthermore, metabolomic analysis revealed that several toxic metabolites, such as puromycin, were enriched in *S. algae*, while many nucleotides were significantly enriched in OP50. Moreover, we found that the “two-component system” was enriched in *S. algae*, whereas “purine metabolism” and “one-carbon pool by folate” were significantly enriched in *E. coli* OP50. Collectively, our data provide new insights to decipher how diet modulates animal fitness and biology.

## 1. Introduction

The study of microbe–host interactions is crucial for understanding animal biology and evolution [1,2]. It has been reported that different habitat bacteria have varying effects on nematode physiology [3,4]. However, the mechanisms underpinning how bacterial food affects nematode fitness and physiology remain elusive.

The diet of marine organisms can have profound effects on their overall physiology. It has been reported that in marine invertebrates like sea urchins, diet composition is linked to significant changes in metabolic pathways, including lipid metabolism, which in turn affect animal growth, reproduction, and stress responses [5,6,7]. It has been shown that the postlarvae of American lobster (*Homarus americanus*) fed with zooplankton are significantly larger and heavier than animals fed with brine shrimp [8]. In addition, it has been described that dietary bile acids significantly promote development, antioxidant capacity, immunity, and intestinal health in abalone (*Haliotis discus hannai*) [9]. It has been reported that dietary inclusion of 4.1% *Clostridium autoethanogenum* protein enhance immunity and disease resistance in *H. discus hannai*, while higher levels (16.25%) reduce oxidative stress resistance [10]. Additionally, a diet containing 41% protein significantly promotes the growth of Australian hybrid abalone (*H. rubra* × *H. laevigata*) at warmer temperatures [11]. Comprehensive omics studies, including transcriptomic and metabolomic analyses, have been conducted to understand how diets influence gene expression profiles and metabolic changes related to nutrient absorption, immune responses, and developmental processes in marine organisms. For example, dietary supplementation with *Bacillus velezensis* significantly promotes the growth and survival rate and increases the expression of genes encoding superoxide dismutase and serine proteinase in Pacific white shrimp, *Litopenaeus vannamei* [12]. Dietary 5-aminolevulinic acid has been reported to significantly promote the growth and immunity of *L. vannamei* by increasing the expression of genes associated with antioxidant immune functions [13]. It has been reported that dietary glycerol monolaurate promotes shrimp growth and lipid metabolism [14]. Another study revealed that astaxanthin feeding, upregulated amino acid, and energy metabolism in the muscles of *Exopalaemon carinicauda* enhanced antioxidant capacity and promoted ATP and unsaturated fatty acid production [15]. These findings underscore the importance of diet as a key factor in the regulation of development and growth, as well as the metabolic and gene expression networks in marine animals, highlighting the need for a deeper understanding of diet-induced physiological and molecular changes.

*Litoditis marina*, a widely distributed free-living bacterivore marine nematode, is an emerging model organism in marine biology [4,16,17]. *L. marina* has a short life cycle, multiple inbred lines with clear genetic backgrounds, high-quality genome assemblies, including annotations and functional genomics resources, and can be easily cultivated in laboratory conditions [16,17]. We characterized the composition of the *L. marina* habitat microbial communities and the natural microbiome-mediated functions on *L. marina* growth and lifespan and identified that CoQ_10_, heme *b*, acetyl-CoA, and acetaldehyde promoted *L. marina* development, while vitamin B6 attenuated nematode growth [4].

*Shewanella algae* is a mesophilic marine bacteria with a worldwide distribution [18] and plays an important role in the turnover of inorganic material by reducing Fe(III) during anaerobic respiration [19,20,21]. *S. algae* is an emerging opportunistic human pathogen and has been implicated in various human disease cases, including skin and soft tissue infections, otitis media, biliary tract infections, vertebral discitis, and bloodstream infections [22,23,24,25,26,27,28,29]. We recently reported that the development and lifespan of *C. elegans* and *L. marina* were significantly attenuated by feeding *S. algae* compared with *Escherichia coli* OP50, the standard laboratory food for *C. elegans* [4]; however, the underlying mechanisms remain incompletely understood.

To further explore the mechanisms underlying the distinct physiological effects of *S. algae* on *L. marina*, compared with *E. coli* OP50 feeding, a combinatorial multi-omics approach was deployed to investigate how different bacterial foods affect the development and aging of the marine nematode *L. marina*. Our results provide novel insights into the molecular and metabolic mechanisms underlying how bacterial food modulates animal fitness and physiology.

## 2. Results

### 2.1. S. algae Delayed Development and Shortened Lifespan of L. marina

Our previous study found that *S. algae* in 2216E media as a food source significantly attenuated *L. marina* development and promoted aging compared with feeding with *E. coli* OP50 [4]. In the present study, we evaluated the development and lifespan of *L. marina* fed with *S. algae* and *E. coli* OP50 in LB media. In line with our previous study, we found that *S. algae* in LB media significantly slowed *L. marina* development in comparison to *E. coli* OP50 (Figure 1A). After 5 days post-hatching, 76% of animals reached the L4 larvae stage when fed with *E. coli* OP50, whereas only 28.67% of animals reached the L4 stage when fed with *S. algae* (Appendix A), which is in accordance with our previous report that 37.62% developed to L4 stage when fed with *S. algae* in 2216E media (*t*-test *p*-value = 0.1286) [4]. In addition, we observed that 82.67% of *L. marina* developed into the L4 stage larvae when fed with *E. coli* OP50, whereas only 48.67% reached the L4 stage when fed with *S. algae* on Day 10 (Appendix A), which is also similar to the result when fed with *S. algae* in 2216E media (42.86%, *t*-test *p*-value = 0.1801) [4]. In accordance with our previous report, we found that *S. algae* feeding promoted *L. marina* aging in comparison to *E. coli* OP50 (Figure 1B). *L. marina* can survive up to 22 days on *E. coli* OP50, with an average lifespan of about 16 days, in contrast to the maximum lifespan of 15 days and an average lifespan of 8 days when fed with *S. algae* in LB media (Appendix A), which is in line with our previous report that the average lifespan was 7 days when fed with *S. algae* in 2216E media (*t*-test *p*-value = 0.3837) [4].

### 2.2. Transcriptomic Analysis of L. marina Feeding with S. algae versus E. coli OP50

To identify the transcriptional characteristics of *L. marina* growing on *S. algae* versus *E. coli* OP50, we performed Illumina RNA sequencing on synchronized L1 worms. Principal component analysis (PCA) showed a clear separation of *L. marina* between feeding with *S. algae* and *E. coli* OP50 (Figure 2A). In total, we identified 703 differentially expressed genes (DEGs), among which 286 were upregulated and 417 were downregulated in *L. marina* fed with *S. algae* compared with *E. coli* OP50 (Figure 2B, Appendix A).

Through GO enrichment analysis, we found that upregulated genes in *L. marina* growing on *S. algae* were significantly enriched in terms such as iron ion binding (*C08G9.2*, *cyp-29A2*, *cyp-14A5*, *cyp-14A2*, *dach-1*, *agmo-1*, *cyp-25A2*, *cdo-1*, *cyp-42A1*, *cyp-36A1*, *drd-1*, and *haao-1*), oxidoreductase activity, acting on paired donors, with the incorporation or reduction of molecular oxygen (*C08G9.2*, *cyp-29A2*, *cyp-14A5*, *cyp-14A2*, *dach-1*, *cyp-25A2*, *fmo-1*, *C01H6.4*, *cyp-42A1*, *cyp-36A1*, and *fmo-2*), oxidoreductase activity (*hacd-1*, *C08G9.2*, *cyp-29A2*, *cyp-14A5*, *daao-1*, *cyp-14A2*, *dach-1*, *agmo-1*, *cyp-25A2*, *fmo-1*, *C01H6.4*, *C15B12.1*, *tdo-2*, *hsd-2*, *cdo-1*, *T10H10.2*, *ldp-1*, *cyp-42A1*, *cyp-36A1*, and *acox-1.1*), and heme binding (*C08G9.2*, *cyp-29A2*, *cyp-14A5*, *cyp-14A2*, *dach-1*, *cyp-25A2*, *tdo-2*, *cyp-42A1*, and *cyp-36A1*) (Figure 2C; Appendix A).

When GO analysis was applied to the downregulated DEGs, we observed that terms such as extracellular region part (*zmp-3*, *ttr-30*, *tep-1*, *ttr-8*, *ttr-59*, *ttr-7*, *ttr-32*, *zmp-4*, *ttr-15*, *test-1*, *ttr-27*, and *ttr-46*), structural constituent of cuticle (*col-141*, *col-68*, *col-142*, *col-166*, *col-117*, *col-73*, *col-176*, *kcc-3*, *col-130*, *col-34*, *col-155*, and *sqt-2*) and structural molecule activity (*col-141*, *col-68*, *col-142*, *col-166*, *col-117*, *col-73*, *col-176*, *kcc-3*, *col-130*, *rpl-35*, *col-34*, *mrpl-51*, *col-155*, *let-2*, and *sqt-2*) were significantly enriched in worms fed with *S. algae* compared with *E. coli* OP50 (Figure 2D; Appendix A).

Additionally, through KEGG analysis, we observed that upregulated DEGs in *L. marina* growing on *S. algae* were significantly enriched in pathways such as tryptophan metabolism (*hacd-1*, *tdo-2*, *alh-11*, *nkat-3*, and *haao-1*) and fatty acid metabolism (*hacd-1*, *pod-2*, *acox-1.1*, *elo-6*, *elo-5*, and *fat-6*); *acox-1.1*, *elo-6*, *elo-5*, and *fat-6* are involved in biosynthesis of unsaturated fatty acids, while *acox-1.1*, *elo-6*, and *elo-5* are essential for fatty acid elongation (Appendix A; Appendix A). When KEGG analysis was applied to the downregulated DEGs, none of the pathways were significantly enriched.

### 2.3. Transcriptomic Analysis of S. algae versus E. coli OP50

Since feeding with *S. algae* attenuated the development and shortened lifespan of *L. marina* compared with *E. coli* OP50 as a food source, we further investigated the transcriptomic characteristics of *S. algae* versus *E. coli* OP50 (Appendix A). Among a total of 1806 single copy ortholog genes between *S. algae* and *E. coli* OP50, we identified 1299 DEGs, with 660 showing upregulation and 639 displaying downregulation in *S. algae* compared with *E. coli* OP50 (Appendix A; Appendix A). From BioCyc functional enrichment, no pathway was enriched in upregulated DEGs. As for the downregulated DEGs, we found that amino acid metabolism-related pathways were significantly enriched in *S. algae* compared with *E. coli* OP50, such as the superpathway of histidine, purine, and pyrimidine biosynthesis (*deoB*, *nrdD*, *gmk*, *hisG*, *hisD*, *hisC*, *prs*, *hisB*, *purH*, *hisI*, *hisF*, *purD*, *hisA*, *prs*, *hisH*, *atpF*, *guaA*, *nrdB*, *pyrH*, and *purK*), the superpathway of L-lysine, L-threonine, and L-methionine biosynthesis I (*thrA*, *thrC*, *thrB*, *metH*, *metL*, *aspC*, *dapA*, *dapF*, *metE*, *metB*, *dapD*, and *serC*), the superpathway of L-threonine metabolism (*ilvB*, *tdh*, *adhE*, *ilvC*, *pflB*, *kbl*, *ilvD*, *ilvA*, *pta*, and *ackA*), and the L-arginine biosynthesis I pathway (via L-ornithine) (*argG*, *gabT*, *argE*, *argC*, *argB*, *argH*, and *argA*) (Figure 3A; Appendix A). Interestingly, genes in the ubiquinol-8 biosynthesis (prokaryotic) pathway (*ubiD*, *ubiE*, *ubiF*, *ubiH*, and *ubiA*), which has been reported to be positively correlated with *L. marina* growth [4], were significantly downregulated in *S. algae* in comparison to *E. coli* OP50 (Figure 3A; Appendix A).

Through GO enrichment analysis, we observed that upregulated genes in *S. algae* were significantly enriched in terms related to RNA processing, such as ncRNA processing (*rluC*, *tilS*, *truC*, *rlmF*, *rsmF*, *selU*, *rlmL*, *thiI*, *tsaD*, and *rlmN*), tRNA processing (*tilS*, *truC*, *selU*, *thiI*, *tsaD*, *rlmN*, *dusC*, *tsaE*, *cmoB*, and *rne*), tRNA wobble base modification (*mnmE*, *tadA*, *mnmA*, *gluQ*, *cmoA*, *selU*, *tusD*, *tusC*, *tilS*, *mnmC*, *tusE*, *cmoB*, and *cmoB*), and ncRNA metabolic process (*rluC*, *tilS*, *truC*, *rlmF*, *rsmF*, *selU*, *rlmL*, *thiI*, *cysS*, and *selA*) (Figure 4A; Appendix A). When GO analysis was applied to the downregulated DEGs in *S. algae*, we found that terms such as alpha-amino acid biosynthetic process (*ilvB*, *tyrA*, *thrA*, *thrC*, *tdh*, *thrB*, *metH*, *glyA*, *hisG* and *gltB*), cellular amino acid biosynthetic process (*ilvB*, *aroF*, *tyrA*, *thrA*, *thrC*, *tdh*, *thrB*, *metH*, *glyA*, and *hisG*), alpha-amino acid metabolic process (*ilvB*, *tyrA*, *dadA*, *thrA*, *thrC*, *tdh*, *thrB*, *metH*, *gcvP*, and *glyA*), and cellular amino acid metabolic process (*ilvB*, *aroF*, *tyrA*, *dadA*, *thrA*, *thrC*, *tdh*, *thrB*, *metH*, and *gcvP*) were significantly enriched (Figure 4B; Appendix A).

Through KEGG annotation, we observed that upregulated DEGs in *S. algae* were significantly enriched in pathways such as transfer RNA biogenesis (*tilS*, *truC*, *selU*, *thiI*, *cysS*, *tsaD*, *dusC*, *tsaE*, *cmoB*, and *mnmA*), and bacterial secretion system (*gspD*, *gspE*, *gspF*, *gspK*, *gspJ*, *gspH*, *secA*, *gspG*, *epsM*, and *gspI*) (Figure 5A; Appendix A). Consistent with the GO analysis, downregulated DEGs in *S. algae* were enriched in several amino acid metabolism process pathways, such as amino acid metabolism (*aroF*, *tyrA*, *ilvB*, *dadA*, *thrA*, *katG*, *puuA*, *glmS*, *thrC*, and *tdh*), glycine, serine, and threonine metabolism (*thrA*, *thrC*, *tdh_1*, *thrB*, *gcvP*, *glyA*, *metL*, *pssA*, *kbl*, and *ilvA*), histidine metabolism (*hisG*, *hisD*, *hisC*, *hisB*, *hisI*, *hisF*, *hisA*, and *hisH*), and cysteine and methionine metabolism (*thrA*, *metH*, *metL*, *aspC*, *gshB*, *sseA*, *mdh_1*, *speE*, *metE*, and *metB*) (Figure 5B; Appendix A).

We subsequently focused on the genes apart from the single copy ortholog genes between *S. algae* and *E. coli* OP50, and selected the top 10% of genes with the highest median absolute deviation (MAD) of expression for downstream analysis, for *S. algae* and *E. coli* OP50, respectively (Figure 3B,C). Through GO enrichment analysis, we observed that terms such as response to xenobiotic stimulus (*prpB*, *acnD*, *prpC*, *nqrD*, *sirA*, *gbpA*, and *DPLKCNFD_00060*) and energy derivation by oxidation of organic compounds (*prpB*, *acnD*, *prpC*, *nqrD*, and *fdhE*) were significantly enriched in *S. algae* (Figure 3B; Appendix A). For the top 10% of genes with the highest MAD of expression in *E. coli* OP50, we observed that GO terms such as sugar transmembrane transporter activity (*exuT*, *rbsB*, *gatZ*, *rbsC*, *rbsA*, *rbsD*, and *kdgT*), NAD(P)H dehydrogenase (quinone) activity (*wrbA*, *nuoK*, *nuoJ*, *nuoH*, and *nuoE*) and oxidoreductase activity, acting on NAD(P)H, quinone, or a similar compound as acceptor (*wrbA*, *nuoK*, *nuoJ*, *nuoH*, *nuoE*, and *qorA*), were significantly enriched (Figure 3C; Appendix A).

KEGG analysis of the selected top 10% of genes revealed that pathways such as propanoate metabolism (*acs*, *prpB*, *acnD*, *DPLKCNFD_02174*, *prpC*, *mmsA*, *pdhA*, and *bkdA2*) and biofilm formation-pseudomonas aeruginosa (*hcpA*, *DPLKCNFD_03118*, *DPLKCNFD_00897*, *DPLKCNFD_01441*, and *ntrC*) were significantly enriched in *S. algae*, while the beta-Lactam resistance pathway (*oppB*, *oppA*, *btuD*, *oppD*, *oppC*, *yejF*, *ompC*, and *ompF*) was significantly enriched in *E. coli* OP50 (Appendix A; Appendix A).

### 2.4. Metabolic Profiling of S. algae- and E. coli OP50-Conditioned Media

LC-MS-based untargeted metabolomic was conducted to profile the metabolic characterization of *S. algae* versus *E. coli* OP50. PCA revealed that media conditioned by *S. algae* and *E. coli* OP50 were separated from each other, and also from the unconditioned media (Figure 6A). We found that 275 differentially accumulated metabolites (DAMs) were significantly enriched by at least 1.5-fold in *S. algae*-conditioned medium, while 157 metabolites were significantly enriched by at least 1.5-fold in *E. coli* OP50-conditioned medium (Figure 6B; Appendix A). In the *S. algae*-conditioned medium, N-(5-Aminopentyl)acetamide, puromycin, 5-hydroxymethyluracil, and thymidine 3′,5′-cyclic monophosphate were among the most enriched metabolites, showing a log_2_ fold enrichment above 5 (Figure 6B; Appendix A). On the other hand, the levels of various nucleotides were significantly decreased in the *S. algae*-conditioned medium, including GDP, TDP, dGMP, UDP-galactose, GMP, CMP, GTP, 3′-AMP, UMP, AMP, cGMP, TMP, ADP, guanine, uracil, 5′-AMP, UDP, and UMP, which were enriched in the *E. coli* OP50-conditioned medium (Figure 6B; Appendix A).

In the *S. algae*-conditioned medium, the most enriched DAMs belonged to lipids and lipid-like molecules (25.47%), organic acids and derivatives (23.58%), organoheterocyclic compounds (22.17%), nucleosides, nucleotides, and analogues (8.49%), and organic oxygen compounds (7.55%) (Figure 6C). While the most enriched DAMs in the *E. coli* OP50-conditioned medium included lipids and lipid-like molecules (37.96%), nucleosides, nucleotides, and analogues (25.93%), organoheterocyclic compounds (12.96%), organic acids and derivatives (11.11%), and benzenoids (5.56%) (Figure 6D). The percentages of organic acids and derivatives, organoheterocyclic compounds, and organic oxygen compounds were more enriched in the *S. algae*-conditioned medium; by contrast, the percentages of lipids and lipid-like molecules, nucleotides, and analogues were more enriched in the *E. coli* OP50-conditioned medium (Figure 6C,D).

Through KEGG pathway-based enrichment analysis on the DAMs between the *S. algae*- and *E. coli* OP50-conditioned media, we found that KEGG pathways such as citrate cycle (TCA cycle) (alpha-ketoglutaric acid, cis-aconitic acid, and citric acid) and glyoxylate and dicarboxylate metabolism (cis-aconitic acid, 4-hydroxy-2-oxoglutaric acid, citric acid, and serine) were significantly enriched in the *S. algae*-conditioned medium (Figure 6E; Appendix A), while pathways including purine metabolism (2,6-dihydroxypurine, adenosine 5′-monophosphate, adenylsuccinic acid, guanosine monophosphate, guanine, dGDP, cGMP, adenosine diphosphate ribose, and adenine) and pyrimidine metabolism (UMP, cytidine-5′-monophosphate, dUDP, dUMP, dTDP, thymidine 5′-monophosphate, and uracil) were significantly enriched in the *E. coli* OP50-conditioned medium (Figure 6F; Appendix A).

### 2.5. Joint Pathway Analysis of DAMs and DEGs in S. algae versus E. coli OP50

To further integrate the DEGs and DAMs in *S. algae* versus *E. coli* OP50, a Joint Pathway Analysis was performed using MetaboAnalyst 6.0 [30]. Joint Pathway Analysis identified 100 and 95 altered pathways in the upregulated and downregulated DEGs and DAMs in *S. algae*, respectively, with five and eight of them showing significant changes (Figure 7A,B; Appendix A). We found that the “two-component system” (TCS) was the most significant upregulated pathway in *S. algae* compared with *E. coli* OP50 (Figure 7A), which is an essential prerequisite for many bacterial pathogenicities [31]. Furthermore, “alanine, aspartate and glutamate metabolism” showed the highest impact score in *S. algae*. In contrast, “pyruvate metabolism”, “glycine, serine and threonine metabolism”, “purine metabolism”, and “one-carbon pool by folate” were significantly enriched in *E. coli* OP50 (Figure 7B).

STITCH interaction analysis revealed that the transcript-metabolite interactions consisted of 104 nodes, which were connected by 381 edges (Figure 7C; Appendix A). Significant interaction was observed only in the enriched metabolites in *S. algae*. Alpha-ketoglutaric acid, ornithine, L-lysine, and N-acetyl-L-ornithine exhibited the highest number of associations (*n* ≥ 6) (Figure 7C). Of note, alpha-ketoglutaric acid was significantly connected to lysine, ornithine, N-acetyl-L-ornithine, and 23 DEGs (*aspC*, *gltB*, *gdhA*, *serA*, *gabT*, *serC*, *sucC*, *argD*, *sucD*, *alaA*, *gabD*, *ppc*, *sad*, *glnA*, *ppsA*, *lysC*, *thrA*, *gltD*, *argC*, *metL*, *dapD*, *nadB*, and *argE*) (Appendix A).

## 3. Discussion

Bacteria can modulate animal fitness and physiology through bacterially-produced metabolites and RNA [32,33]. For example, it has been reported that bacterial siderophore enterobactin promotes *C. elegans* iron acquisition and development [34]. Zhang et al. identified 244 bacterial mutants that attenuated *C. elegans* development, with several of the causal bacterial genes encoding the *bo* oxidase of the electron transport chain and iron transporters [35]. In addition, bacterial peptidoglycan muropeptides have been described to support the development of *C. elegans*, by reducing mitochondrial oxidative stress and enhancing ATP synthase activity [36]. Different bacterial diets, such as *Methylobacterium braciatum, Xanthomonas citri*, and *Sphingomonas aquatilis*, could uniquely alter the development, longevity, and transcriptomic characteristics of *C. elegans*, highlighting the impact of bacterial diet on animal physiological outcomes [37]. In addition, bacterial polysaccharide colanic acid has been identified to extend *C. elegans* lifespan by modulating mitochondrial homeostasis [38]. It has been reported that *E. coli* mutants with reduced levels of methylglyoxal biosynthesis, promote *C. elegans* longevity via suppression of TORC2/SGK-1 and induction of DAF-16 [39]. Like model nematode *C. elegans*, many marine nematodes utilize bacteria as their primary food source; however, how bacterial diet affects the development and aging of marine nematode *L. marina* remains largely unexplored.

### 3.1. Higher Expression of Genes in Iron Ion Binding and Oxidoreductase Activity Pathways Might Promote Aging of L. marina When Grown on S. algae

For the upregulated DEGs in marine nematode *L. marina* grown on *S. algae* compared with *E. coli* OP50, several genes in GO terms, such as iron ion binding (*agmo-1*, *cdo-1*, and *haao-1*), oxidoreductase activity (*agmo-1*, *tdo-2*, *cdo-1*), and heme binding (*tdo-2*) were particularly interesting (Figure 2C; Appendix A). *agmo-1* encodes alkylglycerol monooxygenase for ether-linked lipids degradation, and *agmo-1* mutants exhibit bacterial infection resistance and contain ether-linked (O-alkyl chain) lipids in comparison to exclusively ester-linked (O-acyl) lipids in wild-type animals, which might provide a more resilient cuticle for *agmo-1* mutants [40]. *cdo-1* is predicted to enable cysteine dioxygenase activity and ferrous iron binding, with its high expression resulting in the shortened lifespan of *C. elegans* [41]. *haao-1* encodes 3-hydroxyanthranilic acid (3HAA) dioxygenase (HAAO), and its knockdown extends the lifespan of *C. elegans* and promotes healthy aging [42]. It has been reported that RNAi knockdown of *tdo-2*, which encodes tryptophan 2,3-dioxygenase (TDO), extends the lifespan and ameliorates neurodegenerative pathology in *C. elegans* [43]. Given that RNAi knockdown of *cdo-1*, *haao-1*, and *tdo-2* extend the lifespan and promote healthy aging in *C. elegans*, we speculated that the higher expression of *cdo-1*, *haao-1*, and *tdo-2* might contribute to the shortened lifespan in *L. marina*.

### 3.2. Decreased Expression of Extracellular Structural Components-Related Genes Might Delay the Development of L. marina fed with S. algae

For the downregulated DEGs in *L. marina* fed on *S. algae* compared with *E. coli* OP50, we observed that terms such as extracellular region part and structural constituent of cuticle were significantly enriched in worms when fed with *S. algae* in comparison to *E. coli* OP50 (Figure 2D; Appendix A). Among the extracellular region part pathway, eight genes were identified as transthyretin-like family genes (TTLs), including *ttr-46*, *ttr-15*, *ttr-7*, *ttr-8*, *ttr-32*, *ttr-27*, *ttr-59*, and *ttr-30* (Appendix A). *C. elegans* TTLs exhibited sensitivity to a variety of environmental stressors, such as reactive oxygen species (ROS) stress, exposure to pathogens, and osmotic imbalances [44,45,46,47]. Among the structural constituent of the cuticle pathway, most genes encode collagens, such as *col-141*, *col-68*, *col-142*, *col-166*, *col-117*, *col-73*, *col-176*, *col-130*, *col-34*, and *col-155* (Appendix A). Collagens are the primary components of nematode cuticles, which are shed to allow for growth when animals molt, and protect the nematode as a physical barrier [48]. RNAi knockdown of *col-141* expression has been reported to promote aging in long-lived *daf-2* and *eat-2* mutants [49]. Thus, the downregulation of genes in extracellular structural components-related pathways might impair nematode cuticle structure and function, and delay the development of *L. marina* when fed with *S. algae*.

### 3.3. Downregulated Bacterial Amino Acid Metabolism Genes in S. algae Might Delay Development and Shorten the Lifespan of L. marina

Through BioCyc functional enrichment of downregulated DEGs in *S. algae* compared with *E. coli* OP50, we found that amino acid metabolism-related pathways were significantly enriched, such as the superpathway of histidine, purine, and pyrimidine biosynthesis (*deoB*, *nrdD*, *gmk*, *hisG*, *hisD*, *hisC*, *prs*, *hisB*, *purH*, *hisI*, *hisF*, *purD*, *hisA*, *prs*, *hisH*, *atpF*, *guaA*, *nrdB*, *pyrH*, and *purK*), the superpathway of L-lysine, L-threonine, and L-methionine biosynthesis I (*thrA*, *thrC*, *thrB*, *metH*, *metL*, *aspC*, *dapA*, *dapF*, *metE*, *metB*, *dapD*, and *serC*), the superpathway of L-threonine metabolism (*ilvB*, *tdh*, *adhE*, *ilvC*, *pflB*, *kbl*, *ilvD*, *ilvA*, *pta*, and *ackA*), and the L-arginine biosynthesis I pathway (via L-ornithine) (*argG*, *gabT*, *argE*, *argC*, *argB*, *argH*, and *argA*) (Figure 3A; Appendix A).

Most of these downregulated genes in *S. algae* versus *E. coli* OP50 contribute to amino acid biosynthesis, which indicates that *E. coli* OP50 feeding might provide more abundant amino acids, promoting the development of *L. marina*, compared with using *S. algae* as the food source [50]. It has been reported that individual supplementation with 18 of the 20 essential amino acids could increase *C*. *elegans* longevity [51]. Additionally, most of the amino acid pool sizes increased in the long-lived *C. elegans* [52]. Proline has been found to prolong the longevity of *C*. *elegans* by triggering a temporary surge in ROS production, which originates from the mitochondrial electron transport chain [53]. Similarly, elevated tryptophan levels can also boost longevity by silencing an enzyme responsible for tryptophan breakdown, resulting in a longer lifespan [54]. Our GO and KEGG analysis also revealed that genes in several amino acid biosynthesis terms or pathways, such as histidine, L-threonine, L-ornithine, arginine, polyamine, glycine, serine, threonine, cysteine, and methionine, were significantly downregulated in *S. algae* compared with *E. coli* OP50, indicating that *E. coli* OP50 might supply more abundant amino acids to promote the development and extend lifespan of *L. marina* compared with using *S. algae* as the food source.

### 3.4. Higher Expression of Virulence-Related Genes in S. algae Might Contribute to the Delayed Development and Shortened Lifespan of L. marina

For the upregulated genes in *S. algae*, we observed that GO terms such as ncRNA processing, tRNA processing, tRNA wobble base modification, and ncRNA metabolic process were significantly enriched (Figure 4A; Appendix A). It has been reported that *E. coli* ncRNA DsrA promotes *C. elegans* aging, through suppressing the expression of diacylglycerol lipase encoding gene *dagl-2* [55]. *mnmA* is involved in tRNA modification and could impact protein synthesis, influencing the expression of virulence factors [56,57]. *selU* has been reported as a virulence gene in the enterotoxin gene cluster [58]. *rlmN*, a key methyltransferase encoding gene, can alter ribosomal RNA to regulate bacterial translation efficacy, affecting traits such as virulence, infectiousness, organismal health, and resilience to environmental pressures [59]. The increased expression of *mnmA*, *selU*, and *rlmN* in *S. algae* might explain the developmental delay and shortened lifespan of *L. marina*.

For the top 10% of genes with the highest MAD of expression in *S. algae*, we found that several genes, such as *sirA*, *gbpA*, *hcpA*, and *ntrC*, might be related to bacterial virulence. For example, *sirA* acts as a positive regulator for the SPI1 pathogenicity island, encoding a Type III secretion system that injects effector proteins directly into the host cell’s cytoplasm [60]. GbpA, a recognized virulence factor, encoded by *gbpA*, was reported to facilitate bacterial colonization by interacting with GlcNAc residues in mucin, which upregulates their activity cooperatively and ultimately produces virulence [61]. *hcpA* has been reported to encode HcpA subunits in Type IV pili (TFP), which are essential for virulence in several Gram-negative bacteria [62]. Furthermore, the *ntrC* gene is crucial for nitrogen assimilation, stress resilience, and bacterial virulence [63]. We thus propose that these aforementioned upregulated virulence genes of *S. algae* might induce growth retardation and aging of *L. marina*.

### 3.5. Impact of Toxic Metabolites and Nucleotide on L. marina Development and Lifespan

We observed that several toxic metabolites, which potentially impair *L. marina* development and lifespan, were significantly enriched in the *S. algae*-conditioned medium, such as puromycin (73-fold) and enrofloxacin (28-fold) (Figure 6B; Appendix A). Puromycin has been reported to interfere with the function of ribosomes and block protein synthesis in both eukaryotes and prokaryotes and is toxic to *C. elegans* in both liquid and solid cultures [64]. Enrofloxacin induces oxidative stress by stimulating peroxyl radicals (H_2_O_2_) and lipid peroxidation [65,66].

It is known that nucleotides are essential for nematode cellular function and development, acting as a crucial nutrient source [67,68]. Previous studies have shown that the supplementation of some nucleotides, such as uridine, thymine, cytidine, orotate, β-aminoisobutyrate, and pyrimidine intermediates, extends *C. elegans* lifespan [69]. We supposed that nucleotides enriched in the *E. coli* OP50-conditioned medium benefit *L. marina* physiology.

It has been reported that the glyoxylate cycle plays a crucial role in the pathogenicity of *Mycobacterium tuberculosis* and is essential for the full virulence of *Candida albicans* [70]. Additionally, a complete TCA cycle has proven to be indispensable for the full virulence of serovar Typhimurium strain SR-11; the deletion of various genes involved in the TCA cycle has been reported to lead to reduced virulence [71]. According to our KEGG analysis, the TCA cycle and glyoxylate and dicarboxylate metabolism pathways were significantly enriched in the *S. algae*-conditioned medium, which might contribute to the developmental delay and short lifespan of *L. marina* when fed on *S. algae*. On the other hand, we observed that the purine and pyrimidine metabolism pathways were significantly enriched in the *E. coli*-conditioned medium, which could explain why *E. coli* OP50 promoted *L. marina* development and longevity compared with feeding with *S. algae*.

### 3.6. “Two-Component System” and “One-Carbon Pool by Folate” Pathways Might Modulate L. marina Physiology

Through Joint Pathway Analysis, we found that the “two-component system” (TCS) was the most significant pathway among the upregulated pathways in *S. algae* compared with *E. coli* OP50. TCS has been reported as a vital signal transduction mechanism in bacteria, participating in a multitude of gene regulatory systems that adapt to fluctuating environmental conditions [31]. Thus, TCS is crucial for pathogenic bacteria’s ability to efficiently adapt to diverse microenvironments both inside and outside their hosts, facilitating their pathogenicity [31]. A previous study showed that two-component sensor kinase *KdpD* (7.4-fold change in *S. algae* versus *E. coli* OP50) played a crucial role in the pathogenesis of *Salmonella typhimurium* to *C. elegans* through the TCS pathway [72]. Our data suggest that the upregulation of the TCS pathway in *S. algae* might enhance its virulence and impair *L. marina* physiology. In contrast, the “one-carbon pool by folate” pathway showed the largest impact score among the downregulated pathways in *S. algae*. It was described that metformin repressed the folate metabolism of *E. coli*, which then modulated methionine metabolism in *C. elegans*, and extended worm lifespan [73]. In addition, it was reported that the one-carbon folate cycle is involved in several long-lived *C. elegans* mutants, and reduced 5 methyl tetrahydrofolate (5MTHF) extends the nematode lifespan [74]. Thus, upregulation of “two-component system” and downregulation of “one-carbon pool by folate” might explain why *S. algae* attenuated *L. marina* development and promoted aging, compared with feeding with *E. coli* OP50.

## 4. Materials and Methods

### 4.1. Worm and Culture

The wild strain of the marine nematode *L. marina* was isolated from intertidal sediments in Huiquan Bay, Qingdao [75]. *L. marina* used in this study was a 23rd generation inbred line (F23) that was cultivated through successive full-sibling crosses within our laboratory, and its cultivation follows previous publications [4,16]. In brief, *L. marina* strains were maintained on seawater nematode growth media (SW-NGM). For regular maintenance, SW-NGM was supplemented with *E. coli* OP50 as their diet, and the worms were cultured at 20 °C.

### 4.2. Bacteria

*E. coli* OP50 was obtained from the *Caenorhabditis* Genetics Center (CGC, https://cgc.umn.edu/, accessed on 19 August 2024). *S. algae* was isolated from Huiquan Bay, Qingdao, China, and stored at 80 °C as glycerol stocks [4]. Both bacteria were stamped out fresh cultures from glycerol stocks onto a rectangular LB plate and then incubated overnight at 37 °C. The colonies on the plate were then used to inoculate in LB media.

### 4.3. Development and Lifespan Assays

To assess the influence of *S. algae* and *E. coli* OP50 on *L. marina* development and lifespan, we examined the growth rate and lifespan as previously reported [4].

### 4.4. Sample Collection

The synchronized L1 larvae fed with *S. algae* and *E. coli* OP50 were collected as previously reported [76]. *L. marina* F23 worms cultured on SW-NGM plates were permitted to lay eggs overnight at 20 °C. Once a substantial quantity of eggs had been deposited on the culture plates, adult worms were harvested by gently rinsing the plates with sterilized seawater. Following this, the eggs situated on the plates underwent a thorough rinse into a collection tube and were subjected to bleaching utilizing Worm Bleaching Solution (comprising Bleach: 10M NaOH: H_2_O in a ratio of 4:1:10) to procure sterile eggs, which were subsequently incubated in sterilized seawater overnight, hatching into synchronized L1 larvae. The synchronized L1 larvae were placed onto plates containing *S. algae* or *E. coli* OP50 as a food source, with three replicates for each treatment. Following 4 h of feeding, conditioned media were washed off using M9 solution, followed by the rapid separation and collection of both nematodes and conditioned media. Animal bodies and conditioned medium were frozen in liquid nitrogen for approximately half an hour before being stored at −80 °C and processed separately.

### 4.5. RNA Library Preparation, RNA Sequencing and Transcriptional Analysis of L. marina

Total RNA extraction was performed using Trizol (Invitrogen, Carlsbad, CA, USA). Then, six RNA libraries, three biological replicates for each bacterial feeding, were constructed using NEBNext^®^ UltraTM RNA Library Prep Kit for Illumina^®^ (New England Biolabs, Ipswich, MA, USA), following the manufacturer’s guidelines. Subsequently, we sequenced the RNA libraries on an Illumina NovaSeq 6000 platform, producing 150 bp paired-end reads.

Quality control and adapter trimming were performed on raw reads using fastp [77]. The minimum base score of Q20 was above 98.87%, and Q30 was above 96.95%. Resulting clean reads were mapped to *L. marina* Genome v1.0 [16] using HISAT2 v2.1.0 (--no-discordant, --no-mixed) [78]. Mapped reads were quantified using featureCounts [79] from Subread v2.0.3 with *L. marina* Geneset v1.0 as annotation (-B, -p). Differential gene expression analysis was conducted with R package DEseq2 v1.38 [80]. DEGs with an adjusted *p*-value < 0.05 and a fold change > 1.3 were regarded as significantly differentially expressed genes. GO and KEGG enrichment analysis was performed using R package clusterProfiler v4.6.2 [81] and Geneset v1.0.

### 4.6. RNA Library Preparation, RNA Sequencing, and Transcriptional Analysis of Bacteria

Microbe-conditioned media were washed off using M9 solution, as described above, and were frozen in liquid nitrogen for approximately half an hour before being stored at −80 °C. Library preparation and RNA sequencing were also performed as described above. For differentially expressed gene (DEG) analysis, only counts from single-copy ortholog genes were used and gene symbols of *E. coli* OP50 were adopted as the representative gene symbols for downstream analysis. Differential gene expression analysis was conducted with R package DEseq2 v1.38 [80]. DEGs with an adjusted *p*-value < 0.05 and a fold change > 1.5 were regarded as significantly differentially expressed genes. Enrichment analysis was performed on KOBAS 3.0 [82].

Protein sequences and gene annotation of *S. algae* and *E. coli* OP50 were obtained from Prokka [83] outputs. Reciprocal best hit search was performed using MMseq2 easy-rbh with default parameters [84]. The non-redundant result of MMseq2 easy-rbh output was used as single-copy orthologs between *S. algae* and *E. coli* OP50.

In addition, counts of genes apart from single-copy ortholog genes were normalized to FPKM using DEseq2 v1.38 [80]. The top 10% of these genes were selected using median absolute deviation (MAD); 230 and 237 genes were selected for *S. algae* and *E. coli* OP50 respectively, which were further enriched using clusterProfiler v4.6.2 [81] for KEGG enrichment analysis and TBtools [85] for GO enrichment analysis. All proteins of *E. coli* OP50 and *S. algae* were annotated using eggNOG-mapper version 2.1.6 [86]. Gene Ontology annotations were extracted from the results under the name of GOs.

### 4.7. Metabolomics

The overnight cultures of *E. coli* and *S. algae* in LB medium were diluted 1:100 in LB medium and incubated at 37 °C for 4–5 h until they reached an OD600 of 0.4. The bacterial samples were flash-frozen in liquid nitrogen for 30 m before being transferred to a −80 °C freezer. Five replicates of bacteria-conditioned media and three replicates of unconditioned LB medium were collected. The samples underwent freeze-drying and were reconstituted using prechilled 80% methanol with a through vortex. Afterward, the samples were left to incubate on ice for 5 min and then centrifuged (15,000× *g*, 4 °C for 15 min). A portion of the supernatant was further diluted to a final concentration comprising 53% methanol using LC-MS grade water. The resultant solution was then transferred to a fresh Eppendorf tube and subjected to another round of centrifugation (15,000× g, 4 °C for 15 min). Finally, the supernatant was injected into the LC-MS/MS system for analysis.

UHPLC-MS/MS analyses were conducted utilizing a Vanquish UHPLC system (ThermoFisher, Dreieich, Germany), seamlessly coupled with an Orbitrap Q ExactiveTM HF-X mass spectrometer (Thermo Fisher). Samples were injected onto a Hypersil Gold column (100 × 2.1 mm, 1.9 μm) employing a 12-min linear gradient at a flow rate of 0.2 mL/min. For the positive and negative polarity modes, eluent A (0.1% FA in water) and eluent B (methanol) were employed, respectively. The solvent gradient was meticulously programmed as follows: starting at 2% B for 1.5 min, transitioning to 2–85% B for 3 min, then ramping up to 85–100% B for 10 min, and finally reverting to 2% B for 12 min. Operation of the Q ExactiveTM HF mass spectrometer was optimized for both positive and negative polarity modes, featuring a spray voltage of 3.5 kV, a capillary temperature of 320 °C, a sheath gas flow rate of 35 psi, an auxiliary gas flow rate of 10 L/min, an S-lens RF level of 60, and an auxiliary gas heater temperature of 350 °C.

The Thermo RAW files were processed using Compound Discoverer v3.3 (ThermoFisher) to execute peak alignment, peak picking, and quantitation for each metabolite. We established the primary parameters as follows: correcting peak area with the first QC, maintaining an actual mass tolerance of 5 ppm, ensuring a signal intensity tolerance of 30%, and using a minimum intensity threshold. Following this, we normalized peak intensities against the total spectral intensity. Next, leveraging the normalized data, we employed predictive methods to discern molecular formulas based on additive ions, molecular ion peaks, and fragment ions. Subsequently, we cross-referenced these peaks with mzCloud, mzVault, and the MassList database to secure precise qualitative and relative quantitative measurements. Annotations were performed using KEGG, HMDB, and LIPIDMaps database. The metabolites with adjusted *p*-value < 0.05 and fold change > 1.5 were considered to be differential metabolites. KEGG enrichment pathways were analyzed using Metaboanalyst 6.0, and only used metabolite sets containing at least 3 entries [30].

### 4.8. Bacterial Transcriptome and Metabolism Conjoint Analysis

Integrated transcriptomic and metabolic analysis was carried out using the Joint Pathway Analysis module of MetaboAnalyst v6.0 [30]. Both metabolic (adjusted *p*-value < 0.05, fold change > 1.5) and transcriptomic (adjusted *p*-value  < 0.05, fold change  > 1.5) datasets were utilized for Joint Pathway Analysis. To assess the potential importance of individual molecules within a network, we uploaded the Entrez IDs of DEGs and the names of metabolites, along with their optional fold changes. In the Joint Pathway Analysis module, various parameters were selected: (i) in the pathway database, metabolic pathways (integrated) were chosen; and (ii) in the algorithm selection, the enrichment analysis using the hypergeometric test, topology measure using degree centrality, and the integration method combining queries were applied. Pathways with an adjusted *p*-value less than 0.05 were considered significant. The pathway impact score summarized the normalized topological measure of altered genes or metabolites present in each metabolic pathway, while the −log_10_(adjusted *p*-value) indicated the results of the enrichment analysis.

Furthermore, the interaction network between the key metabolites and DEGs involved in the Joint Pathway Analysis was established using STITCH [87,88], an online tool for visualizing biological relationships between metabolites and genes, as well as between metabolites and metabolites, and genes and genes. A bipartite subgraph was generated by STITCH due to the presence of edges only between proteins and chemicals. The interactions obtained were then used to construct the final differential network image using Cytoscape software v3.10.2 [89].

## 5. Conclusions

In conclusion, we performed a comparative metabolome and transcriptome analysis to uncover the impacts of *S. algae* and *E. coli* on *L. marina* growth and lifespan. We found that *S. algae* potentially impaired *L. marina* development and longevity by downregulating the expression of amino acid metabolism genes, increasing the expression of virulence-related genes, releasing potential toxic metabolites, and reducing nucleotide metabolism. Future research should focus on unraveling the molecular mechanisms underlying how key nematode genes, bacterial genes, and metabolites modulate animal development, aging, and behavior when grown on different bacterial foods, using CRISPR genome editing, RNAi, and diet supplementation with candidate bacterial metabolites.

## Figures and Tables

**Figure 1 ijms-25-09111-f001:**
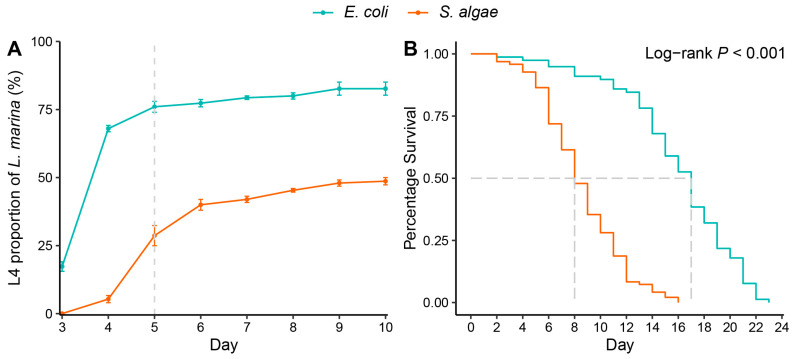
*S. algae* delayed development and shortened the lifespan of *L. marina*. (**A**) *S. algae* significantly attenuated *L. marina* development. *p*-value (Day 5) = 0.00136. *p*-value (Day 10) = 0.00093. *p*-values were calculated with the two-tailed Student’s *t*-test. 70 hatched L1s were transferred onto each conditioned media. The number of L4 worms was scored every day. (**B**) *S. algae* significantly shortened the lifespan of *L. marina*. Log-rank test was applied for the significance.

**Figure 2 ijms-25-09111-f002:**
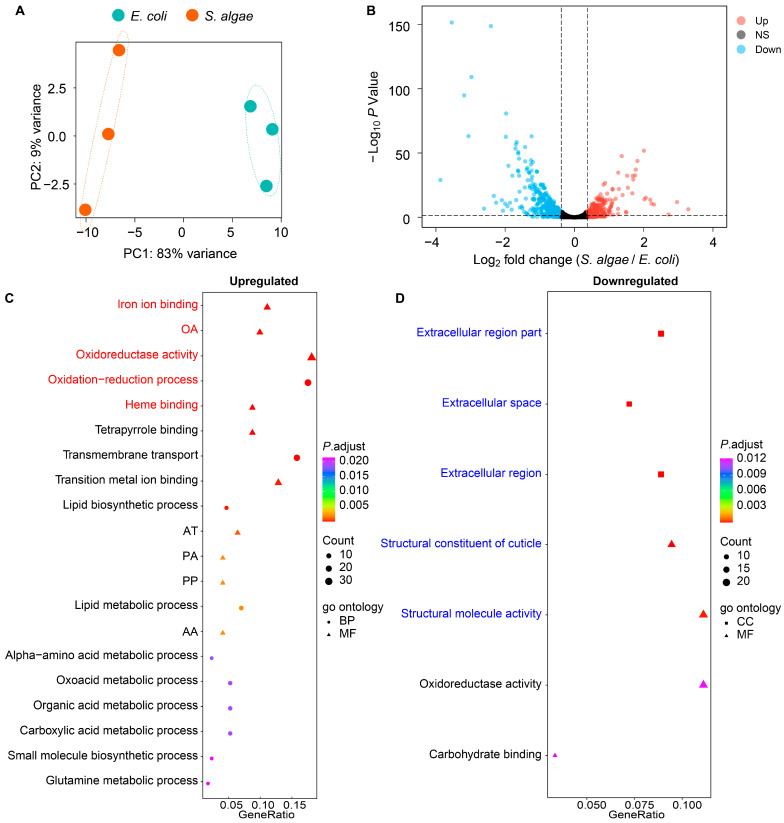
Transcriptional characteristics of *L. marina* growing on *S. algae* versus *E. coli* OP50. (**A**) Principal component analysis (PCA) of *L. marina* gene expression changes growing on *S. algae* and *E. coli* OP50. (**B**) Volcano plot showing differentially expressed *L. marina* genes growing on *S. algae* versus *E. coli* OP50. Up, upregulated genes of *S. algae* vs. *E. coli* OP50; NS, genes with no significant changes; Down, downregulated genes of *S. algae* vs. *E. coli* OP50. (**C**,**D**) GO enrichment analysis for DEGs of *L. marina* growing on *S. algae* versus *E. coli* OP50. BP, biological process; CC, cellular component; MF, molecular function. The color from red to purple represents the significance of the enrichment. GeneRatio is calculated as the ratio of annotated differential genes to the total number of differential genes within a given GO term. OA: oxidoreductase activity, acting on paired donors, with incorporation or reduction of molecular oxygen. AT: active transmembrane transporter activity. PA: primary active transmembrane transporter activity. PP: P-P-bond-hydrolysis-driven transmembrane transporter activity. AA: ATPase activity, coupled to transmembrane movement of substances. The details of GO enrichment analysis are shown in Appendix A.

**Figure 3 ijms-25-09111-f003:**
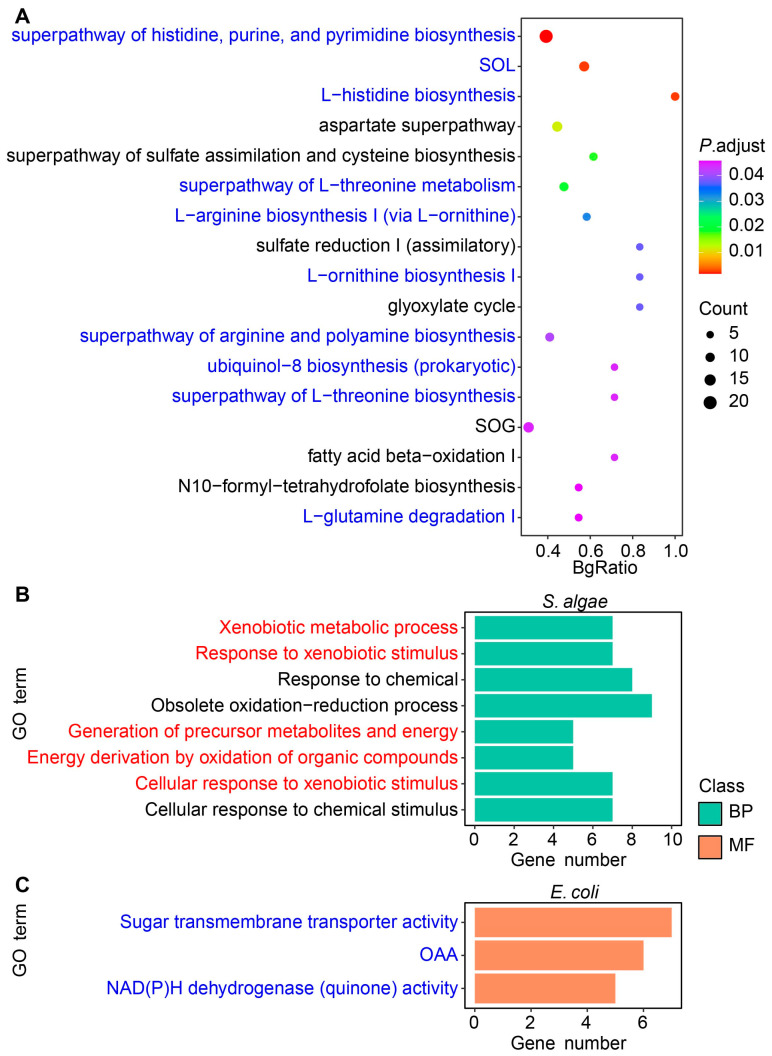
Transcriptional characteristics of *S. algae* versus *E. coli* OP50. (**A**) BioCyc enrichment analysis for downregulated DEGs of *S. algae* versus *E. coli* OP50. The color from red to purple represents the significance of the enrichment. SOL: superpathway of L-lysine, L-threonine, and L-methionine biosynthesis I. SOG: superpathway of glycolysis, pyruvate dehydrogenase, TCA, and glyoxylate bypass. The details of BioCyc enrichment analysis are shown in Appendix A. (**B**,**C**) GO enrichment analysis for the top 10% MAD genes of *S. algae* and *E. coli* OP50, respectively. BP, biological process; MF, molecular function. OAA: oxidoreductase activity, acting on NAD(P)H, quinone, or similar compound as acceptor.

**Figure 4 ijms-25-09111-f004:**
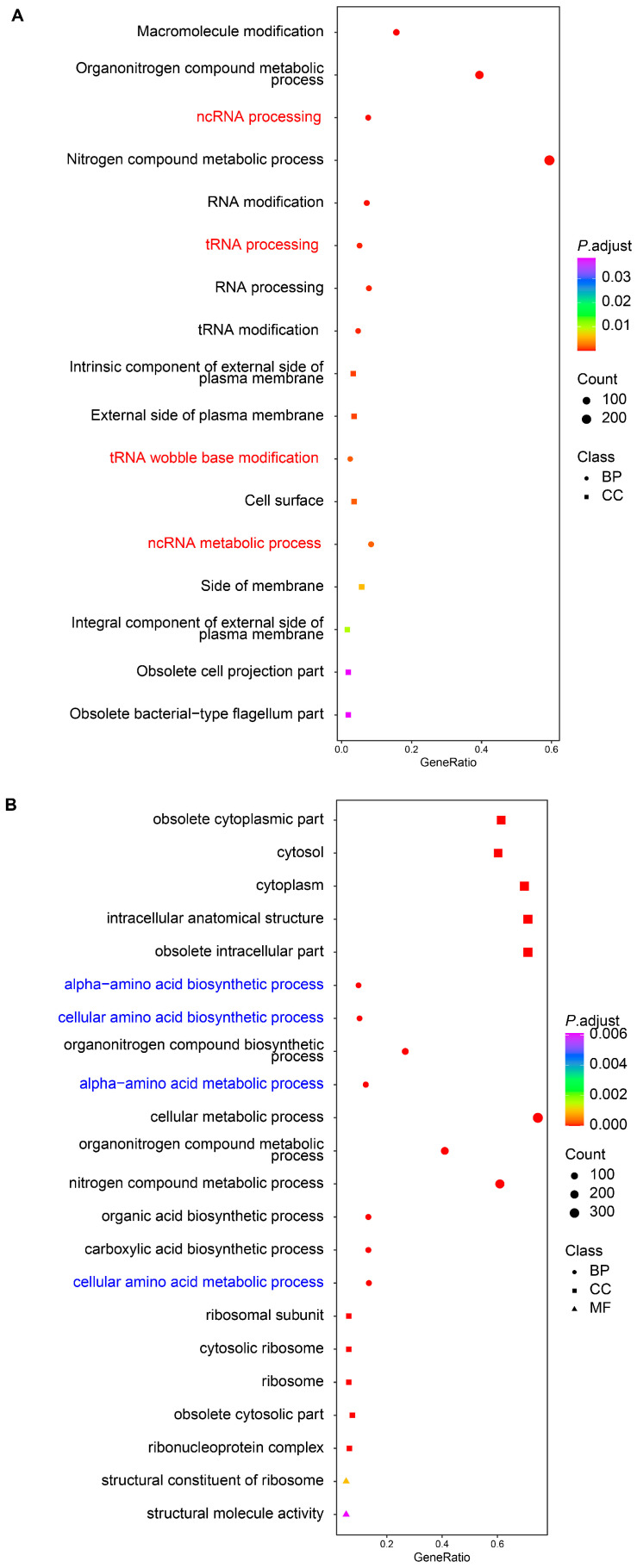
GO enrichment analysis of DEGs in *S. algae* versus *E. coli* OP50. (**A**) GO enrichment of upregulated DEGs in *S. algae* compared with *E. coli* OP50. (**B**) GO enrichment of downregulated DEGs in *S. algae* compared with *E. coli* OP50. The color from red to purple represents the significance of the enrichment. BP, biological process; CC, cellular component; MF, molecular function. Details are shown in Appendix A.

**Figure 5 ijms-25-09111-f005:**
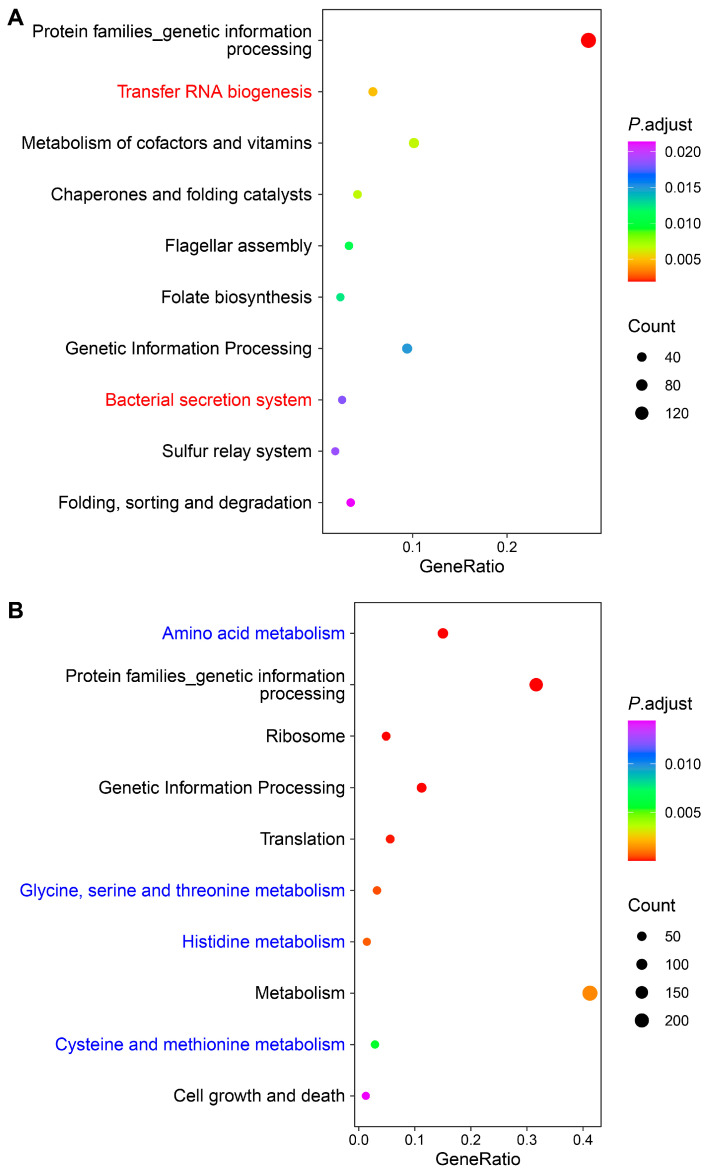
KEGG enrichment analysis of DEGs in *S. algae* versus *E. coli* OP50. (**A**) KEGG enrichment of upregulated DEGs in *S. algae* compared with *E. coli* OP50. (**B**) KEGG enrichment of downregulated DEGs in *S. algae* compared with *E. coli* OP50. The color from red to purple represents the significance of the enrichment. Details are shown in Appendix A.

**Figure 6 ijms-25-09111-f006:**
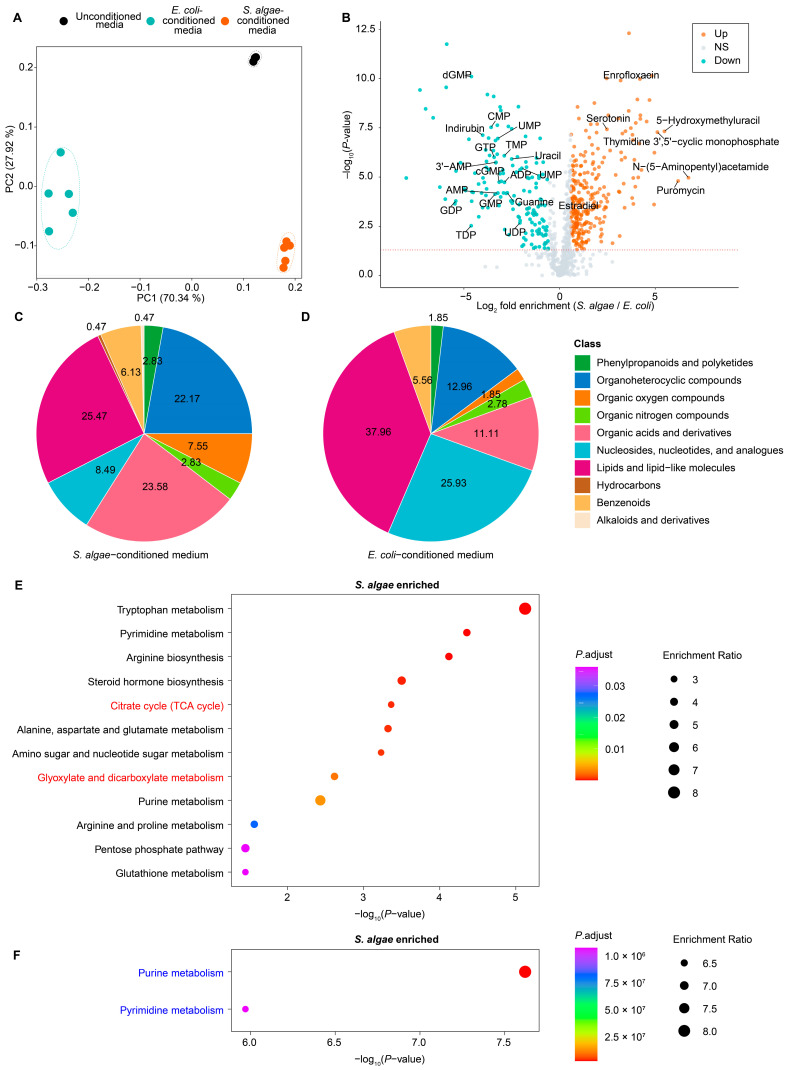
Metabolomics profiling of the *S. algae*- and *E. coli* OP50-conditioned media. (**A**) PCA of the metabolomes of unconditioned medium, *S. algae*-conditioned medium, and *E. coli* OP50-conditioned medium. Five replicates of *S. algae*-conditioned medium and *E. coli* OP50-conditioned medium, as well as three replicates of unconditioned medium were analyzed. (**B**) Volcano plot comparing *S. algae*- and *E. coli*-conditioned media. Log_2_ fold enrichment in *S. algae* over *E. coli* OP50 is plotted on the horizontal axis, and the associated *p*-value is plotted on the vertical axis. The red dashed line shows *p*-value = 0.05. (**C**,**D**) Pie charts show the category of DAMs in *S. algae* (**C**)- and *E. coli* OP50 (**D**)-conditioned media. (**E**,**F**) KEGG enrichment of the significantly changed metabolites. (**E**) Upregulated pathways. (**F**) Downregulated pathways. The color from red to purple represents the significance of the enrichment.

**Figure 7 ijms-25-09111-f007:**
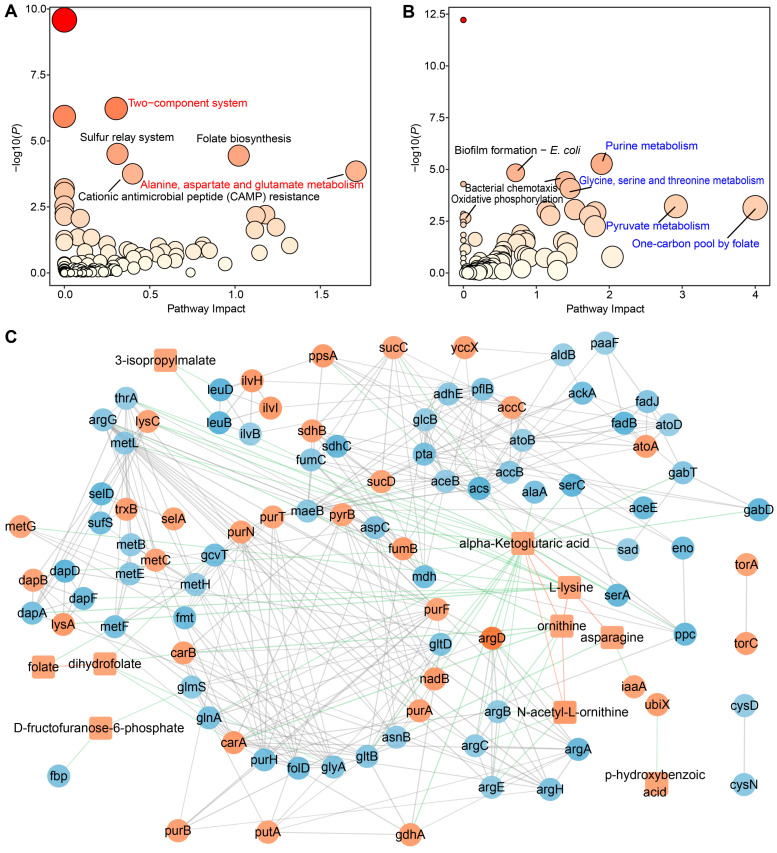
The integrative analysis of transcriptome and metabolome. (**A**) Joint Pathway Analysis between the upregulated DAMs and DEGs in *S. algae* versus *E. coli* OP50. (**B**) Joint Pathway Analysis between the downregulated DAMs and DEGs in *S. algae* versus *E. coli* OP50. Each circle represents a single metabolic pathway, with the size of the circle proportional to the pathway’s impact. The color indicates the pathway’s significance, ranging from highest (red) to lowest (yellow). The enrichment analysis was performed using a hypergeometric test, and the topology measure was assessed by degree centrality. (**C**) The network visualization of STITCH interactions was generated using Cytoscape. Interactions between significant DAMs (square) and DEGs (circle) are displayed, with the edge thickness proportional to the interaction score in the STITCH database. The orange and blue node color indicates the upregulated and downregulated DEGs/DAMs in *S. algae* compared with *E. coli* OP50. Green, red, and gray edges represent interactions between metabolite–gene, metabolite–metabolite, and gene–gene, respectively.

## Data Availability

The datasets used in this study have been deposited in the NCBI Sequence Read Archive under the BioProject ID: PRJNA1138507. Scripts used for custom computational methods are available at: https://github.com/Erique29/Multi-omics-integrative-analysis (accessed on 19 August 2024).

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
