# Peer review of "Multi-Omics Integrative Analysis to Reveal the Impacts of Shewanella algae on the Development and Lifespan of Marine Nematode Litoditis marina"

_ijms, 2024, doi:10.3390/ijms25169111_

Round 1

Reviewer 1 Report

Comments and Suggestions for Authors

The authors conducted a comprehensive study using metabolome and transcriptome analysis to understand the impact of diet on L. marina growth and lifespan. The results were well presented and analyzed. In general, the main area for improvement is the content of introduction and presentation of the figures. Specific comments are lay out below:

·        The introduction should cover current state of the research field. Please add more literature reviews about other studies on how diets affect marine organisms, and research on how their metabolism and transcriptome are altered. The authors mentioned some of these contents in the first paragraph on page 11 in the Discussion section, please consider move some of the content to the Introduction section.

·        In Figure 2B, please also label N-(5-Aminopentyl)acetamide, 5-hydroxymethyluracil, and thymidine 3',5'-cyclic monophosphate as the one you label for puromycin.

·        On line 276, the authors are actually referring to Figure 5C instead of Figure 5B in the text, please update the description accordingly.

There are several details that are also suggested to revise:

·        In Figure 2B, could the authors update the color code for “Up” and “Down” with two different colors other than the same color that are used in Figure 2A representing E. coli and S. algae. It may cause confusion just looking at the figure, but the authors described clearly in the text.

·        Please consider move some of the figures in the supplementary information to the main body, for example, Figure S3 and Figure S4, as they demonstrated important information and were described in both Results and Discussion section.

Comments on the Quality of English Language

The quality of English language is good. This manuscript is well-written and easy to understand.

Reviewer 2 Report

Comments and Suggestions for Authors

The presented research is interesting. Due to the nature of the research, it can be seen that the authors put a lot of work into the preparation and conduct of the research. I have some remarks and questions: - The quality of the 2,4,5 drawings is poor, the descriptions in the drawings are blurred. - Did the author take into account that the observed changes in the development and lifespan of S. algae may be due to differences in the toxins or metabolites they secrete? - Please specify the supplier of the original L. marina strain. - Could the type of medium on which the bacteria were grown affect their subsequent nutritional properties in relation to L. marina? - Please elaborate Conclusion. Our results provide important insights into how S. algae impair the growth and lifespan of L. marina.

Author Response

Thank you very much for taking the time to review this manuscript. Please find the detailed responses below and the corresponding in track changes in the re-submitted files.

Reviewer #2 (Remarks to the Author):

The presented research is interesting. Due to the nature of the research, it can be seen that the authors put a lot of work into the preparation and conduct of the research. I have some remarks and questions:

  1. The quality of the 2,4,5 drawings is poor, the descriptions in the drawings are blurred.

We thank this reviewer for this essential suggestion. We have enhanced the quality of Figures 2, 6, and 7, and included the PDF versions in the supplementary files.

  1. Did the author take into account that the observed changes in the development and lifespan of S. algae may be due to differences in the toxins or metabolites they secrete?

We thank the reviewer for this important comment. Based on our results, we observed that S. algae in both 2216E and LB medium could attenuate L. marina development and lifespan. We then analyzed the toxins and metabolites using metabolomic techniques and discovered that puromycin (73-fold increase in S. algae-conditioned medium) and enrofloxacin (28-fold increase in S. algae-conditioned medium) are likely the key toxins secreted by S. algae. The relative discussion was shown in lines 431–437.

  1. Please specify the supplier of the original L. marina strain.

We thank the reviewer for this important point. The wild strain of the marine nematode L. marina was isolated from intertidal sediments in Huiquan Bay, Qingdao [75], and the L. marina used in this study was a 23rd generation inbred line that was cultivated through successive full-sibling crosses within our laboratory. Accordingly, we added the corresponding description in methods part of the revised manuscript (lines 478–479).

Lines 470–471: “The wild strain of the marine nematode L. marina was isolated from intertidal sediments in Huiquan Bay, Qingdao [75].”

Reference:

[75] Zhang, P.; Xue, B.; Yang, H.; Zhang, L. Transcriptome Responses to Different Salinity Conditions in Litoditis marina, Revealed by Long-Read Sequencing. Genes 2024, 15, 317.

  1. Could the type of medium on which the bacteria were grown affect their subsequent nutritional properties in relation to L. marina?

We thank the reviewer for bringing up these important points. Based on our experiment results, there were no significant difference of L. marina development rate and lifespan between feeding with S. algae in 2216E and LB medium. On day 5, 28.67% of L. marina reached the L4 stage when fed S. algae in LB media, compared to 37.62% that developed to the L4 stage when fed S. algae in 2216E media (t-test P-value = 0.1286). On day 10, 48.67% of worms reached the L4 stage when fed S. algae, which is similar to the 42.86% observed when fed S. algae in 2216E media (t-test P-value = 0.1801). L. marina had an average lifespan of 8 days when fed S. algae in LB media, which is consistent with the 7-day average lifespan observed when fed S. algae in 2216E media (t-test P-value = 0.3837). The corresponding P-values were added in the revised version.

  1. Please elaborate Conclusion. Our results provide important insights into how S. algae impair the growth and lifespan of L. marina.

We thank the reviewer for this important point. As suggested, we elaborate this sentence in lines 608‒612.

Lines 608‒612: “We found that S. algae impaired L. marina development and longevity potentially through downregulating the expression of amino acid metabolism genes, increasing the expression of virulence-related genes, releasing potential toxic metabolites, and reducing nucleotide metabolism.”